# Detection of *Gyrovirus galga 1* in Cryopreserved Organs from Two Commercial Broiler Flocks in Japan

**DOI:** 10.3390/v14071590

**Published:** 2022-07-21

**Authors:** Masaji Mase, Yu Yamamoto, Hiroshi Iseki, Taichiro Tanikawa, Aoi Kurokawa

**Affiliations:** 1National Institute of Animal Health, 3-1-5 Kannondai, Tsukuba 305-0856, Japan; yyu@affrc.go.jp (Y.Y.); hiseki@affrc.go.jp (H.I.); ttanikawa@affrc.go.jp (T.T.); kurokawaa998@affrc.go.jp (A.K.); 2United Graduate School of Veterinary Sciences, Gifu University, 1-1 Yanagido, Gifu 501-1193, Japan; 3Graduate School of Life and Environmental Sciences, Osaka Prefecture University, Izumisano 598-8531, Japan

**Keywords:** *Gyrovirus galga 1*, Japan, PCR

## Abstract

*Gyrovirus galga 1* (GyVg1, previously recognized as avian gyrovirus 2), which was first reported in chicken in 2011, is a new member of the genus Gyrovirus. The presence of GyVg1 has also been confirmed in different regions of Europe, South America, Africa, and Asia, indicating its global distribution. However, because there are no reports of examining the distribution of GyVg1 in animals in Japan, the epidemiology of this virus is unknown. In this study, we attempted to retrospectively detect GyVg1 in cryopreserved chicken materials derived from different two commercial broiler flocks in 1997. The GyVg1 genome was detected in organ materials derived from both flocks by PCR. GyVg1 detected in both flocks was classified into four genetic groups by analyzing the nucleotide sequences of the detected PCR products. These results suggest that diverse GyVg1 strains were present in commercial chicken flocks as early as 1997 in Japan.

## 1. Introduction

*Gyrovirus galga 1* (GyVg1, previously recognized as avian gyrovirus 2) is a new member of the genus Gyrovirus with similarity to chicken infectious anemia virus (CAV). GyVg1 was first detected in the sera of chicken exhibiting clinical signs such as apathy and loss of weight in 2011 in Brazil [1]. The GyVg1 genome is approximately 2.3 kb in length and consists of circular single-stranded DNA, and it mainly comprises three overlapping open-reading frames encoding VP1, VP2, and VP3 [1,2].

Abolnik et al. stated that GyVg1 infections in chickens can result in brain damage, mental decline, and weight loss [3]. Although other specific symptoms of GyVg1 infection have not been observed, autopsy-based studies reported clinical manifestations such as hemorrhage, edema, glandular gastric erosion, and facial and head swelling in infected chickens [3].

Although chicken is considered the natural host of GyVg1 [4], experimental infection in chickens using isolated has have not been conducted because the isolation of virus using cultured cells has not been described. Thus, the pathogenicity to chicken of GyVg1 are still unknown.

GyVg1 has also been detected in different regions of Europe [5], South America [5], Africa [6], and Asia [4,7], indicating its global distribution. In addition, Sauvage et al. [8] identified human gyrovirus, which shares 96% nucleotide identity with GyVg1, from the skin swab of a healthy individual, indicating that GyVg1 might also infect humans. Subsequently, GyVg1 was detected in human blood samples [9,10], human feces [11,12], ferret feces [13], snakes [14], ticks [15], and dogs [16], suggesting that the virus can infect a broad range of animals.

In Japan, since the distribution of GyVg1 in animals including chicken has not been investigated at all to date, the epidemiology of this virus in animals is unknown. In this study, we attempted to retrospectively detect GyVg1 in various organs of chicken obtained from two different commercial broiler farms in 1997. To our knowledge, this study is the first report to show the presence of GyVg1 in Japanese chickens. The GyVg1 genome was detected in the examined samples, and thus, we further conducted detailed genetic analysis.

## 2. Materials and Methods

### 2.1. Cryopreserved Chicken Organ Samples

At two commercial broiler flocks (Farms M and N) in Japan in 1997, sampling was periodically performed four times for routine viral surveillance (Table 1 and Table 2). Significant clinical signs were not observed in these flocks. Major organs, such as the trachea, lungs, liver, spleen, kidneys, pancreas, and rectum, were collected from five birds, and 10–20% tissue homogenates were prepared from these organs. These homogenates were stored at −80 °C at our laboratory.

### 2.2. Detection of GyVg1 by PCR

The homogenates were then centrifuged at 3000× *g* for 10 min. Total DNA was extracted from each organ homogenate using a QIAamp DNA Micro Kit (Qiagen Inc., Valencia, CA, USA) following the manufacturer’s instructions. The DNA was resuspended in nuclease-free water and used as the template for PCR.

PCR to detect GyVg1 was conducted using GyVg1-specific primers (F, 5′-CGTGTCCGCCAGCAGAAACGAC-3′; R, 5′-GGTAGAAGCCAAAGCGTCCACGA-3′; nucleotides 656–1001). These primers were previously designed from the highly coserved region of the GyVg1 genome, and they amplify a 346-bp fragment and target a genomic region that encodes part of the VP2 and VP3 genes [7]. The PCR mixture contained 5 μL of 10× PCR buffer, 0.5 μL of each primer (20 pM), 1.25 U of Ex Taq DNA polymerase (Takara Bio Co., Ltd., Shiga, Japan), 4 μL of 10 mM dNTP, 1 μL of the sample DNAs, and sterilized Milli-Q water to a total volume of 50 μL. The PCR protocol consisted of 35 cycles of 94 °C for 30 s, 60 °C for 30 s, and 72 °C for 30 s. The PCR products were analyzed by 1.5% agarose gel electrophoresis. The obtained PCR products were purified with Montage (Merck, Darmstadt, Germany) according to the manufacturer’s instructions. The PCR products were sequenced directly in both directions using Sanger sequencing and confirmed to be derived from GyVg1 by BLASTn searches. 

### 2.3. Near-Complete Nucleotide Sequence for Representative GyVg1 Strains

The near-complete genome sequences of GyVg1 for representative detected strains were determined using primer sequences based on previous studies [4,8]. The sequenced fragments were assembled using ATGC-Mac ver.5 (GENETYX CORPORATION, Tokyo, Japan). The obtained sequences were deposited in GenBank (accession number: LC716405-8).

### 2.4. Genetic and Phylogenetic Analyses

The obtained nucleotide sequences were analyzed using GENETYX-Mac software (GENETYX CORPORATION, Tokyo, Japan) and compared with known GyVg1 sequences deposited in GenBank. 

We conducted phylogenetic analysis of the nucleotide sequences of GyVg1 with available sequences from GenBank using the ClustalX sequence alignment program, and a phylogenetic tree was constructed by the neighbor-joining method using MEGA7 software [17]. All tools were run with default parameters unless otherwise specified.

## 3. Results

### 3.1. Detection of GyVg1 from Cryopreserved Chicken Organ Samples

The results of GyVg1 detection in the organs examined in this study are presented in Table 1 and Table 2. Figure 1 presents an example of agarose gel electrophoresis of PCR products. The obtained sequences revealed that all PCR products were derived from GyVg1 genomes.

GyVg1 was detected in chicken older than 27 days at both farms (Table 1 and Table 2). The virus was detected in 12 chicken on farm M and 7 chicken on farm N. The organ with the highest detection rate was the liver (18 samples/19 positive birds), followed by the spleen (17 samples/19 positive birds) (Table 1 and Table 2). However, in some samples, GyVg1 was detected in all major organs examined, such as chicken No. 1 and 5 (48 days old) on M farm and chicken no. 1 (44 days old) on N farm. The detection rates of GyVg1 were lower in the trachea and lungs than in the liver and spleen.

### 3.2. Sequence Analysis of GyVg1 Detected in Organ Samples by PCR

By determining the nucleotide sequences of the PCR products derived from each organ, GyVg1 samples detected on farm M were identical (Table 1). Conversely, GyVg1 samples detected on farm N could be divided into three genetic groups (Table 2). Interestingly, two different groups of GyVg1 were detected in 31-day-old chicken, and two other groups were detected in 44-day-old chicken (Table 2). Based on phylogenetic analysis, four different groups were identified and termed G1–4 (Figure 2).

All GyVg1 samples detected in chicken on farm M were type G1. Conversely, regarding samples obtained from 31-day-old chicken on N farm, the GyVg1 sample detected in chicken no. 4 was type G2, and the sample detected in chicken no. 5 was type G3. Furthermore, among 44-day-old chicken, GyVg1 samples detected in chicken Nos. 1–3 were type G4, and samples detected in chicken Nos. 4–5 were type G2 (Table 2).

GyVg1 detected on farm M (G1) displayed 100% nucleotide sequence identity with G13 and HLJ1506-2 in BLAST searches [2,13] (Figure 2, Table 3). GyVg1 detected in the liver of chicken no. 4 (31 days old) on farm N (G2) exhibited 100% nucleotide sequence identity with strains JL1511 and HLJ1508, which were isolated from chickens in China [2]. GyVg1 detected in spleen of chicken no. 5 (31 days old) on farm N (G3) displayed 99.7% nucleotide sequence identity with strains GZ1601 and HLJ1506-1 [2]. GyVg1 detected in the spleen of chicken no. 1 (44 days old) on farm N (G4) exhibited 100% nucleotide sequence identity with strain G17 [13], which was isolated from a ferret in Hungary. Thus, the four genetically different strains of GyVg1 detected on two different farms examined in this study displayed high genetic similarity to foreign strains. Additionally, multiple genotypes existed in the same farm, and multiple genotypes were present in chickens of the same age.

### 3.3. Near-Complete Genome Analysis of Representative GyVg1 Strains

To obtain an in-depth understanding of the genetic characteristics of GyVg1 in Japan, nearly complete genome sequences of selected samples including the VP1, VP2, and VP3 coding regions were determined excluding the GC-rich region of approximately 40 bp in the 3ʹ-untranslated region (UTR). It has been reported that GyVg1 can be classified into different clusters based on the amino acid sequences of VP1, VP2 and VP3 proteins [2].

Via comparisons with GyVg1 nucleotide sequences available in GenBank by BLASTn searches, GyVg1 detected on farm M (G1, representative strain JP/KGSM/M0313-2Li/97) exhibited 99.8% nucleotide sequence identity with strain G13 [13] (Figure 3).

Conversely, GyVg1 detected in the liver of chicken no. 4 on farm N (G2, representative strain JP/KGSM/N0313-4Li/97) exhibited 98.5% nucleotide sequence identity with a strain from chicken meat (GenBank accession no. MT671981). GyVg1 detected in the spleen of chicken no. 5 on farm N (G3, representative strain JP/KGSM/N0313-5S/97) exhibited 98.9% nucleotide sequence identity with strain G17 [13]. GyVg1 detected in the spleen of chicken no. 1 on farm N (G4, representative strain JP/KGSM/N0326-1S/97) displayed 99.8% nucleotide sequence identity with strain G17 [13].

### 3.4. VP1 Amino Acid Sequences of Representative GyVg1 Strains

VP1 is the major viral structural protein of CAV, and it contains an abundant neutralization antigenic epitope and plays a key role in virus growth and transmission [18]. All VP1 amino acids sequences currently reported are 460 amino acids in length [2], including those found in Japan. At present, the reported GyVg1 strains have been classified into three genetic clusters (I–III) by phylogenetic analysis using the VP1 amino acid sequence [2], but the Japanese strains detected in this study were divided into two genetic clusters (I and II) (Table 4, Figure 4A). Strains JP/KGSM/M0313-2Li/97 and JP/KGSM/N0313-4Li/97 classified into cluster I exhibited 100% identity with the HLJ1506-2 strain detected in China. In addition, JP/KGSM/N0313-5S/97 and JP/KGSM/N0326-1S/97 classified into Cluster II exhibited 100% identity with the G17 strain detected in Hungary [13].

### 3.5. VP2 Amino Acid Sequences of Representative GyVg1 Strains

Most reported VP2 amino acid sequences are 231 amino acids in length, and those found in Japan, excluding one strain (JP/KGSM/N0326-1S/97, G4), were also this length. This strain is identical to the G17 strain, which has a serine insertion at amino acid 162 as reported previously. The VP2 sequences were well conserved within their major phylogenetic clusters [2]. The currently reported GyVg1 strains were classified into four genetic clusters (I–IV) by phylogenetic analysis using the VP2 amino acid sequence, and interestingly, the Japanese strains detected in this study were also divided into four clusters (Table 4, Figure 4B). The JP/KGSM/N0326-1S/97 strain was classified into Cluster II similarly as the G17 strain. The sequence of phosphatase motif WLRQCARSHDEICTCGRWRSH (amino acids 95–115) was also conserved in Japanese GyVg1 strains, which indicated the importance of VP2 for GyVg1 [2].

### 3.6. VP3 Amino Acid Sequences of Representative GyVg1 Strains

The VP3 protein in CAV is also known as apoptin, which was demonstrated to induce apoptosis of hematopoietic cells, causing the observed anemia in infected chicken [19]. It was revealed that GyVg1 VP3 protein can also induce apoptosis of tumor cells [20]. Regarding the predicted protein sequences of VP3 of GyVg1, three strains, excluding the JP/KGSM/N0326-1S/97 strain, were 124 amino acids in length. The JP/KGSM/N0326-1S/97 strain had an arginine insertion at amino acid 122, similarly as the G17 strain. Currently reported GyVg1 strains were classified into five genetic clusters (I–V) by phylogenetic analysis using the VP3 amino acid sequence, but interestingly, the Japanese strains detected in this study were divided into four clusters (Table 4, Figure 4C). The aforementioned JP/KGSM/N0326-1S/97 strain was classified into Cluster II similarly as the G17 strain.

### 3.7. Analysis of Direct Repeats (DRs) of Representative GyVg1 Strains

A previous study described novel motif patterns in the DR region in the 5ʹ-UTR [2]. The 5′-UTR of GyVg1 is by definition located between the canonical polyadenylation site (AATAAA) and the transcription site. Although the function of the noncoding region is not yet known, it may have the same promoter-enhancer function as its CAV counterpart, which includes four or five 21-base DRs and an indispensable 12-bp insert. According to these criteria [2], we observed three different types of DR sequences in Japanese GyVg1 strains. G1 was classified as pattern E, G2 was classified as pattern D, and G3 and G4 were classified as pattern B.

## 4. Discussion

The existence of GyVg1 was first confirmed in 2008 in Brazil [1], but the present study revealed that it was already present in commercial broiler flocks in the 1990s in Japan.

In this report, we detected GyVg1 in materials collected periodically at two different farms. The results illustrated that chickens older than 27 days were infected by the virus. GyVg1 was detected in many organs in infected chickens, most commonly in the liver and spleen. This is consistent with findings that GyVg1 in chickens in China is most commonly isolated from the liver and spleen, suggesting that these organs are suitable for detecting this virus.

GyVg1 was not detected in younger chickens, but it becomes detectable as chickens age. Regarding CAV infection in commercial broiler flocks, at 35 days of age, the virus was detectable by PCR in 16 of 20 chickens, and by 42 days of age, the virus was detectable in 18 of 20 chickens [21]. This suggests that the mode of infections of GyVg1 in commercial broiler flocks is similar to that of CAV. Importantly, the existence of multiple genotypes of GyVg1 in the flocks, even at the same farm and in chickens of the same age, indicates that viral surveillance should be conducted using individual samples as opposed to pooled samples.

GyVg1 can be classified into multiple genotypes by phylogenetic analysis based on the amino acid sequences of VP1, VP2, and VP3, but the Japanese GyVg1 strains detected in this study were also classified into multiple genotypes. This indicates that multiple genotypes were already present on Japanese farms in the 20th century. Additionally, the amino acid sequences of VP1, VP2, and VP3 remain well-conserved within their major phylogenetic clusters [2].

To date, the epidemiology, host range, transmission route, and pathogenesis of GyVg1 remain poorly understood. In addition, the pathogenicity of GyVg1 against chicken is unclear. The only report suggesting the pathogenicity of this virus is a case reported in South Africa [3]. Low-pathogenic NDV and GyVg1 were detected in this case, but their involvement, including interactions, was not clarified. The possibility of a synergistic pathogenic effect between avirulent NDV and GyVg1 requires further investigation. Interestingly, as GyVg1 infection in ticks was reported recently, it may also be possible that the virus spreads among flocks via ticks [15]. Furthermore, Varela et al. reported the detection of GyVg1 in poultry vaccines, indicating the potential role of contaminated vaccines in the spread of GyVg1 [22].

Most GyVg1 strains are detected mainly in chickens, but they have been also detected in various hosts such as ferrets, snakes, and dogs, suggesting possible cross-species transmission [7,11,12,13,14,16]. Additionally, GyVg1 has been detected in the blood of healthy humans, transplant recipients and HIV-positive patients [9,10]. Cases of infection apparently caused by the consumption of infected chicken have been reported in humans and snakes.

Therefore, it is important to understand the epidemiology of GyVg1 in chicken; however, little is known about the viral infection or antibody production in chicken. As the genomic information of GyVg1 has been determined, antibody testing using proteins expressed using recombinant gene techniques would be useful for clarifying the infiltration status in animals. However, the isolation of the virus from clinical samples is absolutely necessary to evaluate its pathogenicity in animals. CAV is known to grow in some cell lines such as MDCC-MSB1 cells [23]. To evaluate the pathogenicity of a virus in animals, it is necessary to establish an effective culture system for GyVg1.

In conclusion, this is the first report to show the presence of GyVg1 in Japanese chickens. GyVg1 was detected in chicken materials derived from two different commercial broiler flocks in 1997. These results suggest that GyVg1 was already present in commercial flocks prior to the first report of GyVg1 infection in 2011 in Brazil. Furthermore, multiple genotypes existed in the same farm, and multiple genotypes were present in chickens of the same age. Unfortunately, the current status of GyVg1 infection in chickens in Japan is unknown. The findings obtained in this study should be useful for further epidemiological research of GyVg1 in commercial flocks.

## Figures and Tables

**Figure 1 viruses-14-01590-f001:**
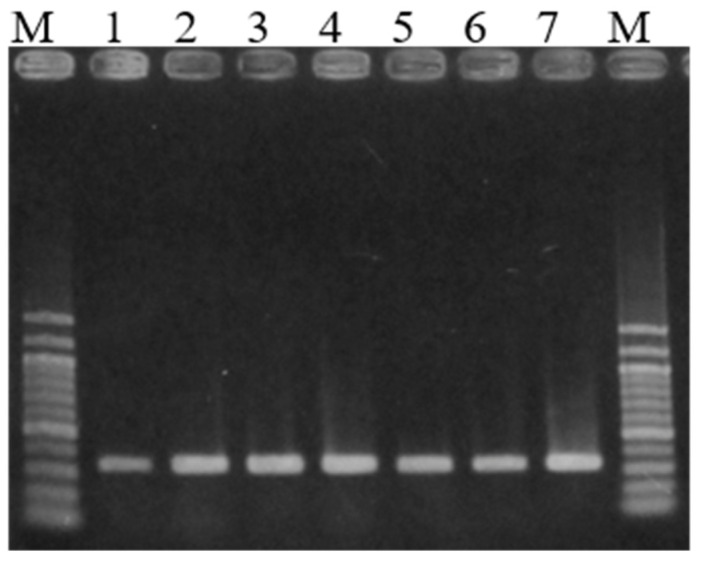
An example of PCR products analyzed by agarose gel electrophoresis. PCR was performed using samples obtained from chicken no. 1 (44 days old) on farm N. M, 100-bp ladder marker; 1, trachea; 2, lung; 3, liver; 4, spleen; 5, kidney; 6, pancreas; 7, rectum.

**Figure 2 viruses-14-01590-f002:**
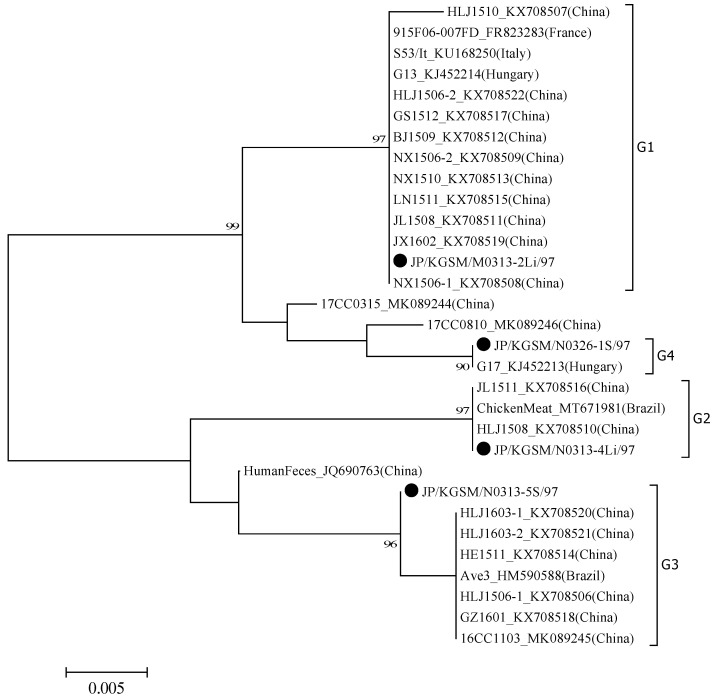
Phylogenetic trees based on the partial gene of GyVg1 strain G13 (GenBank accession no. KJ452214). Nucleotides 678–978 were subjected to phylogenetic analysis. Subsequently, a tree was generated using the neighbor-joining method in MEGA 7 [17] with 1000 bootstrap replicates. All tools were run with default parameters unless otherwise specified. Then, horizontal distances were proportionally set to the minimum number of nucleotide differences required to join nodes and sequences. The Japanese GyVg1 strains are indicated by black circles.

**Figure 3 viruses-14-01590-f003:**
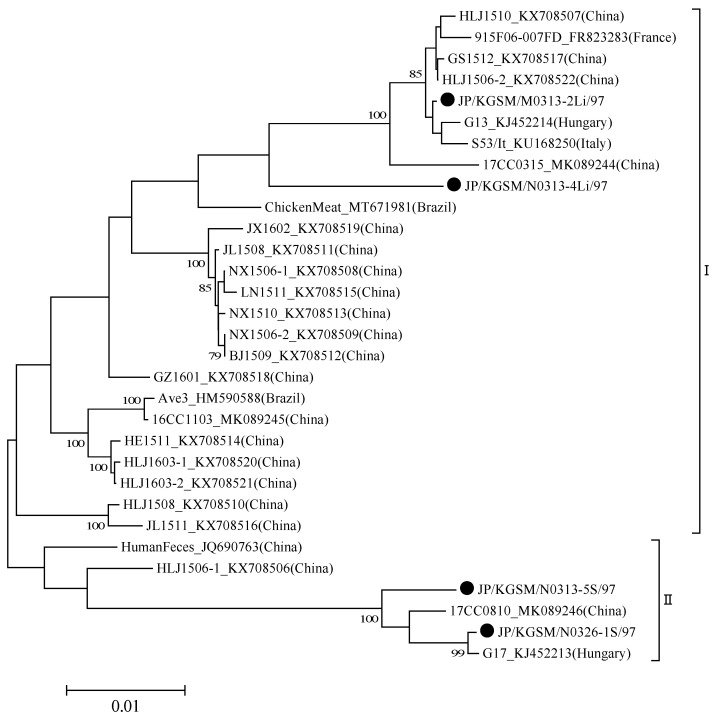
Phylogenetic trees based on the near-complete genome of GyVg1 strain G13 (GenBank accession no. KJ452214). Nucleotides 1–2335 were subjected to phylogenetic analysis. Subsequently, a tree was generated using the neighbor-joining method in MEGA 7 [17] with 1000 bootstrap replicates. All tools were run with default parameters unless otherwise specified. Then, horizontal distances were proportionally set to the minimum number of nucleotide differences required to join nodes and sequences. The Japanese GyVg1 strains are indicated by black circles. The genotypes were defined as described by Yao et al. [2].

**Figure 4 viruses-14-01590-f004:**
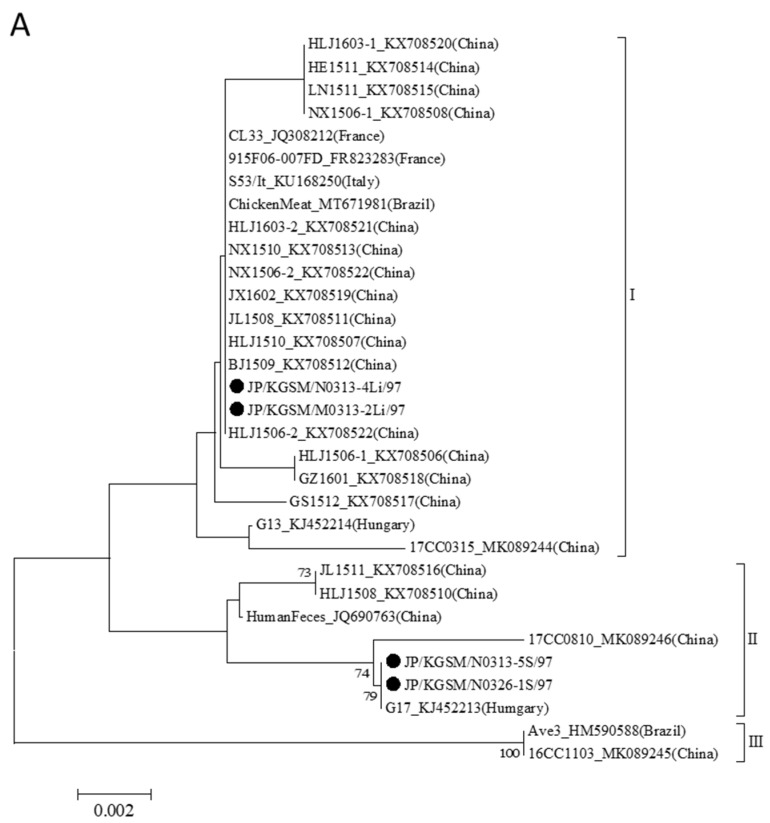
Phylogenetic analysis of the amino acid sequences of VP1 (**A**), VP2 (**B**), and VP3 (**C**) of GyVg1. The predicted amino acid of complete sequences of each protein was subjected to phylogenetic analysis. Subsequently, a tree was generated using the neighbor-joining method in MEGA 7 with 1000 bootstrap replicates. All tools were run with default parameters unless otherwise specified. Then, horizontal distances were proportionally set to the minimum number of nucleotide differences required to join nodes and sequences. The Japanese GyVg1 strains are indicated by black circles. The genotypes were defined as described by Yao et al. [2].

**Table 1 viruses-14-01590-t001:** Detection of GyVg1 in various organs in chicken from farm M.

Age (Days)	Positive Birds/Examined Birds	Chick no.	Trachea	Lung	Liver	Spleen	Kidney	Pancreas	Rectum	Genetic Groups
18	0/5	1–5	-	-	-	-	-	-	-	
		Detection rate	0/5	0/5	0/5	0/5	0/5	0/5	0/5	
27	2/5	1	-	-	-	-	-	-	-	
		2	-	-	-	-	-	-	-	
		3	-	-	+	+	-	-	-	G1
		4	+	+	+	+	+	+	+	G1
		5	-	-	-	-	-	-	-	
		Detection rate	1/5	1/5	2/5	2/5	1/5	1/5	1/5	
35	5/5	1	-	-	+	+	-	-	-	G1
		2	+	+	+ *	+	+	+	+	G1
		3	-	+	+	+	+	+	+	G1
		4	-	+	+	+	+	+	+	G1
		5	-	+	+	+	+	-	-	G1
		Detection rate	1/5	4/5	5/5	5/5	4/5	3/5	3/5	
48	5/5	1	+	+	+	+	+	+	+	G1
		2	+	+	+	+	+	+	+	G1
		3	-	-	+	+	+	-	-	G1
		4	-	+	+	+	+	-	-	G1
		5	+	+	+	+	+	+	+	G1
		Detection rate	3/5	4/5	5/5	5/5	5/5	3/5	3/5	
Total Detection rate	12/20		5/20	9/20	12/20	12/20	10/20	7/20	7/20	

* used as representative strain.

**Table 2 viruses-14-01590-t002:** Detection of GyVg1 in various organs in chicken from farm N.

Age (Days)	Positive Birds/Examined Birds	Chick no.	Trachea	Lung	Liver	Spleen	Kidney	Pancreas	Rectum	Genetic Groups
14	0/5	1–5	-	-	-	-	-	-	-	
23	0/5	1–5	-	-	-	-	-	-	-	
		Detection rate	0/10	0/10	0/10	0/10	0/10	0/10	0/10	
31	2/5	1	-	-	-	-	-	-	-	
		2	-	-	-	-	-	-	-	
		3	-	-	-	-	-	-	-	
		4	-	-	+ *	+	-	-	-	G2
		5	-	-	-	+ *	-	-	-	G3
		Detection rate	0/5	0/5	1/5	2/5	0/5	0/5	0/5	
44	5/5	1	+	+	+	+ *	+	+	+	G4
		2	+	-	+	-	+	+	+	G4
		3	+	-	+	+	+	-	-	G4
		4	+	-	+	-	+	-	-	G2
		5	+	-	+	+	+	+	+	G2
			5/5	1/5	5/5	3/5	5/5	3/5	3/5	
Total Detection rate	7/20		5/20	1/20	6/20	5/20	5/20	3/20	3/20	

* used as representative strain.

**Table 3 viruses-14-01590-t003:** Genetic comparison of nucleotide sequences of PCR products detected in first screening PCR by GenBank searches.

Representative Strain Name	Chicken	GyVg1 Groups in This Study	Viruses with the Highest Homology	Percent Homology
JP/KGSM/M0313-2Li/97	Farm M, 35 days old, no.2 chicken liver	G1	G13(KJ452214), HLJ1506-2 (KX708522)	100
JP/KGSM/N0313-4Li/97	Farm N, 31 days old, no.4 chicken liver	G2	JL1511 (KX708516), HLJ1508 (KX708510)	100
JP/KGSM/N0313-5S/97	Farm N, 31 days old, no.5 chicken spleen	G3	GZ1601 (KX708518), HLJ1506-1 (KX708506)	99.7
JP/KGSM/N0326-1S/97	Farm N, 44 days old, no.1 chicken spleen	G4	G17 (KJ452213)	100

**Table 4 viruses-14-01590-t004:** Summary of classification based on different genetic regions.

Representative Strain Name	Cluster				
	Regions Detected by First Screening PCR	Near Complete Genome	VP1	VP2	VP3
JP/KGSM/M0313-2Li/97	G1	I	I	I	I
JP/KGSM/N0313-4Li/97	G2	I	I	III	III
JP/KGSM/N0313-5S/97	G3	II	II	IV	V
JP/KGSM/N0326-1S/97	G4	II	II	II	II

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
