# Peer review of "Detection of Gyrovirus galga 1 in Cryopreserved Organs from Two Commercial Broiler Flocks in Japan"

_viruses, 2022, doi:10.3390/v14071590_

Round 1

Reviewer 1 Report

1. On page 6, the title of Table 3. "Genetic comparison of nucleotide sequences of detected avian gyrovirus 2 (AGV2) samples ir ? ", the text at the end of the sentence appears to be incomplete.

2. Page 11, Discussion, in lines 248-250, the author mentions, as AGV2 infection in mites was reported recently, it is also be possible that the virus spreads among flocks via mites [15]. But in reference 15, the author analyzes target was ticks, not mites.

Author Response

Responses to Reviewer1 comments on viruses-1808605

   I am pleased to note the favorable comments of the editor and reviewer and have made correction that I hope meet with their approval. We added a column to make the strain names in Table 3 easier for understanding to readers. The changes were highlighted in yellow.

Reviewer 1:

  1. On page 6, the title of Table 3. "Genetic comparison of nucleotide sequences of detected avian gyrovirus 2 (AGV2) samples ir ? ", the text at the end of the sentence appears to be incomplete.

Thank you for your advice. We corrected the title of Table3.

  1. Page 11, Discussion, in lines 248-250, the author mentions, as AGV2 infection in mites was reported recently, it is also be possible that the virus spreads among flocks via mites [15]. But in reference 15, the author analyzes target was ticks, not mites.

Thank you for your advice. We corrected "mite" to "tick".

      I would like to thank the editor and reviewer for their helpful comments, and hope that the revised manuscript is acceptable for publication on Viruses.

Yours sincerely,

Masaji MASE, D.V.M., Ph,D.

Reviewer 2 Report

The authors retrospectively detected AGV2 in cryopreserved chicken materials derived from different two commercial broiler flocks in 1997. The AGV2 genome was detected in organ materials derived from both flocks by PCR. AGV2 detected in both flocks was classified into four genetic groups by analyzing the nucleotide sequences of the detected PCR products. These results suggested that diverse AGV2 strains were present in commercial chicken flocks as early as 1997 in Japan. 

The investigation is interesting. However, Since AGV2 has appeared in 1997, how about the current status of AGV2 infection in chicken flocks in Japan? The authors did not provide any data. 

In addition, AGV2 was renamed as Gyrovirus galgal base on Simona Kraberger's report (Archives of Virology, 2021).

Author Response

Responses to Reviewer 2 comments on viruses-1808605

  I am pleased to note the favorable comments of the editor and reviewer and have made correction that I hope meet with their approval. We added a column to make the strain names in Table 3 easier for understanding to readers. The changes were highlighted in yellow.

Reviewer 2:

  1. The investigation is interesting. However, Since AGV2 has appeared in 1997, how about the current status of AGV2 infection in chicken flocks in Japan? The authors did not provide any data.

  Thank you for your advice. This study is the first report to show the presence of GyVg1 in Japanese chickens. Unfortunately, we do not have other samples that can examine the presence of GyVg1, so the detection of GyVg1 have not performed other than the cryopreserved samples used in this study. Furthermore, there is no report by other researchers investigating the existence of GyVg1 in Japanese chickens. Therefore, the current status of GyVg1 infection in chickens is unknown. We are currently preparing to collect the new samples to investigate the recent GyVg1 infection status in chickens. Additionally, we hope that this report will trigger the research for GyVg1 for chicken samples that have been stored in other research institutes in Japan.

We added some sentences in the manuscript to make this situation easier for the reader to understand (L46-9,L268-275).

  1. In addition, AGV2 was renamed as Gyrovirus galga l base on Simona Kraberger's report (Archives of Virology, 2021).

Thank you for your advice. We changed "AGV2" to " GyVg1(Gyrovirus galga l )".

      I would like to thank the editor and reviewer for their helpful comments, and hope that the revised manuscript is acceptable for publication on Viruses.

Yours sincerely,

Masaji MASE, D.V.M., Ph,D.

Round 2

Reviewer 2 Report

No anything else suggestions.